# Conservation Practices for Personal Protective Equipment: A Systematic Review with Focus on Lower-Income Countries

**DOI:** 10.3390/ijerph20032575

**Published:** 2023-01-31

**Authors:** Cassandra L. Thiel, Pallavi Sreedhar, Genevieve S. Silva, Hannah C. Greene, Meenakshi Seetharaman, Meghan Durr, Timothy Roberts, Rajesh Vedanthan, Paul H. Lee, Gizely Andrade, Omar El-Shahawy, Sarah E. Hochman

**Affiliations:** 1Department of Population Health, NYU Grossman School of Medicine, New York, NY 10016, USA; 2Columbia College, Columbia University, New York, NY 10027, USA; 3Perelman School of Medicine, University of Pennsylvania, Philadelphia, PA 19104, USA; 4Social Science Division, New York University Abu Dhabi, Abu Dhabi P.O. Box 129188, United Arab Emirates; 5College of Literature, Science, and Arts, University of Michigan, Ann Arbor, MI 48109, USA; 6Health Sciences Library, NYU Grossman School of Medicine, New York, NY 10016, USA; 7Department of Oral and Maxillofacial Surgery, NYU Grossman School of Medicine, New York, NY 10016, USA; 8Department of Emergency Medicine, Lewis Katz School of Medicine, Temple University, Philadelphia, PA 19140, USA; 9Department of Medicine, Division of Infectious Diseases and Immunology, NYU Grossman School of Medicine, New York, NY 10016, USA

**Keywords:** PPE, sustainability, waste, resilience, conservation, efficiency

## Abstract

During the start of the COVID-19 pandemic, shortages of personal protective equipment (PPE) necessitated unprecedented and non-validated approaches to conserve PPE at healthcare facilities, especially in high income countries where single-use disposable PPE was ubiquitous. Our team conducted a systematic literature review to evaluate historic approaches for conserving single-use PPE, expecting that lower-income countries or developing contexts may already be uniquely conserving PPE. However, of the 50 included studies, only 3 originated from middle-income countries and none originated from low-income countries. Data from the included studies suggest PPE remained effective with extended use and with multiple or repeated use in clinical settings, as long as donning and doffing were performed in a standard manner. Multiple decontamination techniques were effective in disinfecting single use PPE for repeated use. These findings can inform healthcare facilities and providers in establishing protocols for safe conservation of PPE supplies and updating existing protocols to improve sustainability and overall resilience. Future studies should evaluate conservation practices in low-resource settings during non-pandemic times to develop strategies for more sustainable and resilient healthcare worldwide.

## 1. Introduction

The emergence of SARS-CoV-2 (COVID-19) strained healthcare resources worldwide. Healthcare facilities and hospitals across the world reported severe shortages of supplies, particularly personal protective equipment (PPE) including masks and respirators, gowns, gloves, and eye protection. These shortages were driven by an unprecedented increase in consumption, compounded by factory closures along the global supply chain [1]. As a result, many hospitals and medical centers worldwide developed individual approaches to manage limited supplies based on guidance from the World Health Organization and data from previous outbreaks [2,3,4,5,6,7]. Hospitals and other health facilities enacted conservation practices such as limiting access to PPE stocks and rationing supplies. Novel, often non-validated, techniques emerged early in the pandemic to prolong the lifespans of single-use PPE, including prolonged use, multiple use, and reprocessing of single-use PPE [8].

Given the likelihood of continued supply disruptions in the recovery from the pandemic and the onslaught of climate change [9,10,11], it is imperative to better understand PPE conservation practices that can be safely employed in times of shortage. These practices must not reduce PPE efficacy to levels that increase risk of pathogen exposure [12]. Systematic reviews of PPE conservation that have been published since the emergence of COVID-19 have been narrowly focused on one type of PPE, mainly N95 respirators [13,14,15,16,17,18], and have not commented on practices in developing contexts. Here, we assess the body of knowledge at the start of the pandemic on various PPE conservation and reuse practices.

We hypothesized that low- and middle-income countries may have validated methods for conserving PPE due to regular shortages in supplies. Ideally, all countries should be enabled to meet their supply chain needs, but existing consumption practices from high-income countries could be improved by drawing from the experiences of countries with fewer resources, as has been suggested in the example of cataract surgeries [19,20,21]. We sought to analyze existing literature to assess effective means of conserving and reusing PPE, with implications for the ongoing pandemic and beyond. We anticipate that our analyses will help bolster healthcare system preparedness for future supply challenges, as well as support efforts to promote long-term environmental sustainability [22].

## 2. Methods

### 2.1. Systematic Review

On 9 June 2020, following PRISMA guidelines for systematic reviews [23], a medical librarian (TR) performed searches for studies without language or date restrictions in the LitCOVID, PubMed, Embase, HealthStar, and CINAHL databases. The search included studies describing the efficacy of methods used to conserve or reuse PPE within healthcare settings and comparisons of single-use and reusable PPE. A full description of search terms and inclusion and exclusion criteria can be found in the Appendix A. The citation lists of relevant systematic reviews were screened for additional studies.

Title, abstract, and full text screenings were completed by 2 researchers (GA, PL) using Covidence software [24]. The entire research team iteratively built and tested a data extraction form (see SM for further details). Data extraction from each of the included studies was carried out by four authors (GS, PS, MS, HG), with two randomized to each study; of those two, one was assigned to consolidate the two versions of extracted data into one dataset for each selected study. If needed, consolidators returned to the original study to resolve a discrepancy.

### 2.2. Quality Analysis

Two authors (PS, MS) independently evaluated the quality of each study to determine the validity of its recommendations. The Cochrane collaboration tool was used to assess risk of bias for randomized controlled trials [25,26,27], while the Newcastle–Ottawa quality assessment scale was used to assess risk of bias in cohort and case–control studies [28]. Discrepancies between scores were reconciled by a third author (SH). Due to the lack of a standardized quality assessment tool, the validity of studies that were lab-based, cross-sectional, or descriptive were evaluated by one author (PS), who compiled the limitations listed in each study from the latter study designs as described by the study’s authors.

### 2.3. Data Analysis

Included studies were stratified into three primary analytic domains: (1) the reuse and extended use of PPE manufactured as single-use, (2) methods for disinfecting PPE, and (3) efficacy of PPE by type and mode of pathogen transmission (droplets, aerosols, and fomites) (Figure 1). Reuse was defined as the repeated donning and doffing of the same pieces of PPE; extended use was defined as the practice of wearing the equipment throughout a shift or prolonged period without donning or doffing. Summary statistics were used to evaluate the types of PPE, modes of pathogen transmission, and settings under study, as well as the type of study, metrics used, and outcomes evaluated.

Because this review sought to evaluate conservation methods for resource-limited settings, the human development index (HDI) was used to estimate the relative accessibility of each publication. In order to compare the applicability and global relevance of findings, we used the 2019 United Nations human development index to stratify study settings by relative country development level [29]. The HDI is a summary measure of achievement in health, education, and standard of living. Countries are considered to have a “very high” level of development with HDI of 0.800 or greater; “high” development with HDI of 0.700–0.799; “medium” development with HDI 0.550–0.699; and “low” with HDI < 0.550. 

## 3. Results

### 3.1. Included Studies

After removing duplicates, the initial search yielded 980 studies (Figure 1). After screening, 50 articles met all inclusion criteria and were included for analysis. Eighteen studies assessed reuse or extended use of single-use PPE, 15 studies analyzed PPE disinfection methods, and 19 studies measured the efficacy of PPE, with 2 of the included studies meeting criteria for two categories. These studies were published in 27 different academic journals, with a plurality published by the *American Journal of Infection Control* (*n* = 10 papers) and the *Journal of Occupational and Environmental Hygiene* (*n =* 5). The publication dates ranged from 2008 to 2020, with 30% of included papers published in 2020 (prior to June) and 58% published in 2015 or later.

#### 3.1.1. Geographic Distribution and Funding Sources

The 50 included studies spanned 14 countries, with some studies assessing multiple locations. The United States (45%) and China (20%) were the two most common settings. Forty-nine of the 50 studies originated in countries considered to have “high” or “very high” HDI, and 38 of these were conducted in countries ranked at a “very high” HDI. Three studies included countries with “medium” HDI, and no study was conducted in countries with “low” HDI. The average HDI of all included studies’ national setting was 0.905, more than the worldwide average HDI of 0.713.

Twenty-nine of the 50 studies were federally funded; four received university funding; three credited both federal and university funding; and 14 did not specify funding sources. Of the 23 studies from the United States, 18 were federally funded and five did not specify.

#### 3.1.2. PPE Type, Study Setting, and Primary Outcome

The type of PPE evaluated varied across studies. N95 or elastomeric respirators were the most common type of masks evaluated (*n* = 35, or 70%), followed by surgical masks (*n* = 15, 30%) and cloth masks (*n* = 4, 8%). Four studies discussed improvised masks, and one studied only melt-blown fabric (a component of masks) due to a shortage of PPE. Aside from masks, five studies examined protective eyewear, two studied gloves, and five studied gowns. Two of the gown studies evaluated gown material rather than complete suits. 

Seventeen studies (34%) took place in an in situ or clinical setting, while the remaining 33 were laboratory-based studies (defined as artificially replicating conditions of disease or PPE use, rather than examining an intervention or disease transmission in a clinical setting). Half of the studies included PPE disinfection (*n* = 25), measuring the impact of disinfection on PPE efficacy, PPE degradation, ability to reuse, and ability to effectively disinfect against specific pathogens.

Fourteen studies (28%) focused on fit testing or filtration efficacy of filtering face piece respirators (FFRs) for aerosols, and three papers studied protection from noninfectious contaminants such as chemicals. Seventeen remaining studies examined the use of PPE to protect against one or more infectious diseases, defined as a pathogen in a clinical setting or a (potentially surrogate) virus or bacteria used in a laboratory. Twelve studies measured degradation of PPE, four focused on degradation related to reuse of single-use PPE and nine studies measured the impact of disinfection on degradation of PPE. Of the 15 studies that measured the impact of PPE disinfection on reduction in contamination, eleven measured the effect of disinfection on viruses found on PPE surfaces and four studies focused on the impact of disinfection on bacteria or bacterial spores. Among all included studies, three specifically studied SARS-CoV-2, the virus causing COVID-19 [30,31,32]. The breadth of studies dedicated exclusively to COVID-19 has notably burgeoned as the pandemic has progressed.

### 3.2. Reuse and Extended Use of Disposable PPE

Of the 18 studies examining reuse or extended use of single-use PPE, five focused on in-situ, clinical settings and 13 were lab-based, as shown in Table 1. Overall, studies concluded that reuse can be done safely with some carefully implemented disinfection methodologies (see Appendix A, for description of all disinfection methods tested in the literature).

Many studies concluded that the extended use of eye protection and N95 respirators could be practiced safely with strict adherence to environmental and hand hygiene. For example, in a study evaluating surface contamination of N95 respirators and eye protection, no SARS-CoV-2 was detected by RT-PCR (reverse transcription polymerase chain reaction) from any surface of PPE worn by healthcare providers exiting hospital rooms of patients with COVID-19. No contamination was discovered on healthcare workers’ goggles, N95 respirators, or shoes worn for extended use while interacting with a series of COVID-19 patients for up to 25 min each [30].

A cross-sectional survey of infection control coordinators at hospitals in Vietnam, Pakistan, and China (medium HDI countries potentially more prone to supply shortages), demonstrated that extended use and reuse of cloth masks, surgical masks, and respirators were common practice. Examination of sample masks from participating sites revealed high penetration of particles through reused cloth and surgical masks compared with N95 respirators. This study proposed that respirators remain the most effective option compared with surgical or cloth masks, and that respirators can be reused as long as they are not visibly soiled or damaged [33].

A cluster-randomized trial conducted by MacIntyre et al. [49] found that hospital staff who wore N95 respirators throughout work shifts had fewer cases of clinical respiratory illness compared with staff who wore N95s only for high-risk procedures or when providing direct patient care under infection isolation precautions. Studies also found that N95 respirators were still effective barriers after donning and doffing. A study based in an infectious disease hospital in Brazil found that N95 respirators could be used by healthcare workers during shifts up to 12 hours and reused for up to five days [35]. A separate study found that N95 respirators could safely undergo five uses with repeated donning and doffing while maintaining filtration factors greater than 100 (considered to be effective protection); some respirator models could undergo 20 uses without fit test failure or a drop in filtration factor [37]. In contrast, a pilot study found that tethering devices such as elastic straps from some N95 respirator models could not withstand more than three donning and doffing repetitions before breaking or losing elasticity [41].

### 3.3. PPE Disinfection Methods

Fifteen studies focused on the disinfection of PPE through a variety of techniques (Table 2). Five studies focused on disinfecting bacterial pathogens in lab-based settings. Of the 10 studies focused on viral pathogens, two were in situ studies and eight were lab-based. In studies examining the reduction of contaminants, a 3 log_10_ or 5 log_10_ reduction in viable microorganisms was considered effective disinfection [50,51,52,53,54,55,56,57,58,59]. Overall, safe reuse depended on the specific mask material, with differences in contamination risk based on hydrophobicity or permeability; the type of contamination (bacterial, viral, non-disease-specific); the mode of application of disinfectant; and a range of other factors.

Viscusi et al. [59] demonstrated that moist heat incubation and microwave-generated steam decontamination—methods that can be used in a variety of settings—did not adversely affect efficacy or comfort with N95 respirators. Additionally, studies comparing multiple methods of disinfection found that ultraviolet germicidal irradiation (UVGI), ethylene oxide, vaporized hydrogen peroxide, and bleach did not adversely affect the efficacy of N95 and elastomeric respirators [42]. However, none of these studies assessed the effect of multiple rounds of decontamination on respirator function. In a separate study, higher intensities of UVGI used for decontamination led to a loss of strength and reduced efficacy of N95 respirator material [43]. Salter et al. [44] measured the deposition of toxic residues on filtering facepiece respirators after decontamination and found that the majority of decontamination methods tested did not leave significant residues on respirators. However, decontamination with ethylene oxide left behind toxic residues, and bleach caused discoloration and a lingering odor.

While many of the decontamination methods mentioned above, such as UVGI, ethylene oxide and vaporized hydrogen peroxide, require significant infrastructure and resources, several studies found that heat sterilization, which can be performed using fewer resources, was effective and could be repeated without detrimental effect to PPE. Steam or dry heat sterilization, repeated up to five times, effectively disinfected single-use N95 respirators without degrading the pressure drop, permeability, filtration factor, or penetrance of bacterial spray [45]. A separate study by Liao et al. found that heat sterilization could be repeated up to 50 times without adversely affecting filtration efficacy of the melt-blown fabric used in N95 respirators [46].

In addition to measuring the effect of disinfection on PPE efficacy and degradation, several studies evaluated disinfection methods against specific pathogens. Many of these disinfection methods, such as steam sterilization and chlorine-based disinfectants, can be used in a variety of settings without significant investment of infrastructure. Steam sterilization, either in the form of microwave-generated steam or steam from a household rice cooker, was effective in killing bacteria (*Staphylococcus aureus*) and influenza virus [39,43,44]. In addition, diluted chlorine solution effectively killed bacterial spores on gown material [57].

### 3.4. PPE Efficacy

PPE efficacy was measured most commonly through infection rate (n = 11) or filtration factor (n = 4). All other methods were grouped into a third category, “other metric” (n = 4). The infection rate studies were all conducted in clinical settings whereas the filtration factor and “other metric” studies were conducted in lab-based settings. Overall, N95 respirators were found to be superior to surgical masks in preventing infection from viruses transmited by aerosols but not more effective than surgical masks for pathogens transmitted through bacteria or fomites. Cloth masks were generally less effective than both surgical masks and N95 respirators. Finally, the one study that compared reusable gowns to disposable gowns found disposable gowns more impermeable to respiratory fluid. 

#### 3.4.1. Surgical Masks and N95 Respirators

Studies comparing infection transmission using either N95 respirators or surgical masks yielded varied results, summarized in Table 3. Overall, N95 respirators were more effective than surgical masks in preventing transmission or infection with viruses transmitted via aerosols. Wang et al. [32] evaluated the effect of N95 respirator use on SARS-CoV-2 transmission and found that N95 respirators were more effective than no mask use to prevent infection from SARS-CoV-2. Surgical masks were not evaluated in this study. With respect to prevention of infection with pathogens spread primarily via droplets or fomites (such as influenza or bacteria), N95s were not consistently more effective than surgical masks. Two randomized controlled studies in China found lower rates of clinical respiratory illness among healthcare workers who wore N95 respirators during work shifts compared with workers who wore surgical masks [49,63]. However, these studies found no difference in influenza-like illness or laboratory-confirmed respiratory viruses among groups and did not take into account infection exposure outside of the work environment.

#### 3.4.2. Cloth, Homemade, and Novel Masks

Studies evaluating the efficacy of cloth or homemade masks in preventing the spread of respiratory viruses are also shown in Table 3. In general, cloth or other homemade masks were less effective than surgical masks or N95s in blocking pathogens that are spread via aerosols. There was significant heterogeneity among studies and paucity of data related to droplet or fomite transmission. In an experimental model of respiratory virus transmission, cloth masks were half as effective as surgical masks and 50 times less effective than filtering facepiece respirators (FFP2, an equivalent to N95s) [76]. A randomized study of healthcare workers in Vietnam found that staff wearing surgical masks had lower rates of influenza-like illness and laboratory-confirmed respiratory virus compared with staff wearing cloth masks [64]. Conversely, a more recent study used aerosolized avian influenza virus and found that homemade cloth masks with internal layers of kitchen paper blocked 95% of aerosolized avian influenza virus, similar to the blocking effect of surgical masks and N95 respirators [61]. Overall, studies of novel or homemade masks yielded inferior results to N95s when evaluating aerosol transmission of pathogens. Our review found a lack of studies evaluating the effectiveness of novel PPE for protection from respiratory pathogens spread through respiratory droplets or fomites [72,74,77].

#### 3.4.3. Other, Non-Mask PPE

Only one study evaluated the efficacy of reusable gowns, which could reduce reliance on disposable gowns during shortages and reduce environmental burdens [78]. In this simulation, all participants wearing reusable surgical gowns had evidence of respiratory fluid permeation onto their underlying scrubs (Table 3). After multiple washes, the reusable gowns lost their protection from permeation, while the control participant wearing a disposable gown had no permeation of respiratory fluid [38].

### 3.5. Quality of Included Studies

Results of our quality and bias assessment are shown in SM Appendix A for included RCT studies and SM Appendix A for included cohort and case–control studies. Of the 50 studies reviewed, 36 did not fall into the categories of RCT, cohort, or case–control and thus were not able to be assessed for bias through either the Newcastle–Ottawa or Cochrane tools. Instead, we conducted a qualitative analysis of bias through comparison of the limitations listed in each paper. The majority of these studies took place in a lab setting and tested the efficacy of various decontamination methods.

For these 36 studies, the most commonly mentioned limitations included small sample size, evaluation of decontamination method for only one type of filtering facepiece respirator, absence of assessing fit and filtration efficiency after decontamination, lack of decontamination effect assessment on mask straps or nose pieces, and only evaluating decontamination efficacy against one microorganism. Overall concerns in many of the lab studies were that decontamination processes were not tested in a true healthcare setting, suggesting further research would be needed before implementing changes in medical facilities. Thirteen of these studies (26%) did not list limitations [31,33,34,35,38,39,53,57,60,61,73,75,77].

## 4. Discussion

This literature review was motivated by the COVID-19 pandemic, which overwhelmed healthcare facilities, disrupted supply chains, and caused shortages of single-use PPE in the U.S. and worldwide [79]. We are also concerned about the sustainability of medical systems, including the waste generation from single-use PPE [12,80] and the impact of climate change on medical supply chains [22]. Understanding the state of the literature on the efficacy of various types of PPE and conservation practices in healthcare settings at the early stages of the COVID-19 pandemic can help contextualize how systems’ responses have since developed, thus informing future supply chain management and resilience strategies. Prioritizing effective, reusable PPE practices could reduce excess waste in highly-resourced settings, allow redistribution of supplies to resource-limited settings, and help prepare for future infectious outbreaks.

We found that there are safe and effective ways to reuse N95s in circumstances when protection from aerosols is necessary. However, any reuse inevitably carries the risk of contamination or reduced efficacy. Many factors contribute to the safe reuse of respirators, such as a standardized process for donning and doffing and stable storage conditions. Improper doffing can enable the transfer of contaminants [40], and storage conditions with limited humidity prevent the survival of contaminating microorganisms [39]. When facing a low quantity of available PPE, extended use can limit waste [30,35] and can be done safely with standardized extended use practices that include proper donning and doffing [30,35,40,49]. Further research should focus on developing low-cost, widely accessible PPE for repeated use.

Some of the included studies developed effective disinfection methods that may not be financially or logistically feasible during outbreaks of infection or in low-resource settings, as they may require significant resource investment [31,57,60]. Disinfection methods that required significant infrastructure or cost include vaporized hydrogen peroxide and pulsed xenon ultraviolet light. On the other hand, effective disinfectants using common and inexpensive methods, such as microwave steam bags and steam sterilization using a kitchen rice cooker, are available to healthcare settings with limited resources [36,42,47,52,54,58,61].

Overall, included studies suggest that N95 respirators are superior to surgical masks with respect to preventing spread and acquisition of pathogens that are spread by aerosols. For pathogens that are spread through respiratory droplets and fomites, surgical masks can be as effective as N95 respirators. Several studies found surgical masks to be more effective than cloth masks. There are several potential reasons for this, including the mode of transmission studied (aerosol vs droplet) and variability in materials or fit of cloth and novel masks [61,64,65,66,76]. This suggests that the cost of surgical masks, rather than use or reuse of cloth masks, is worth prioritizing despite limited resources. The additional cost of N95s can be reserved for aerosol-generating procedures, direct care of patients with respiratory pathogens that are transmitted via aerosols, and during seasons with high transmission of respiratory viruses.

Our findings are consistent with those of Rowan and Laffey’s recent rapid review of SARS-CoV-2-related PPE use and reuse, which found varied global techniques for PPE reuse and reprocessing in the setting of the COVID-19 pandemic [8]. They suggest that appropriate disinfection and reuse of face coverings are important public health measures and that uptake of reusable PPE should increase. Similarly, literature reviews of N95s by Rodriguez-Martinez et. al, Seresirikachorn et. al, O’Hearn et. al, Toomey et. al, Paul et. al., and Gnatta et al. [13,14,15,16,18,81] conclude multiple decontamination approaches are appropriate for N95s, most notably ultraviolet germicidal irradiation and vaporized hydrogen peroxide; however, all studies note a lack of strong research.

Our review spanned 14 countries and included analyses conducted both during times of disease outbreak and otherwise. It also captured multiple unconventional and innovative study methodologies. The variety of conclusions in the literature illustrates that responses to infectious disease outbreaks are highly context- and microbe-specific, requiring robust but sustainable supply chains. Standardizing the metrics of PPE efficacy could enable more accurate comparison and determination of best practices for conservation.

### Limitations

Given the variety of study settings and approaches in the included literature, we were unable to quantitatively compare equivalent outcome metrics or variables across all studies. Instead, we qualitatively compared the main findings and recommendations. In addition, our review did not include publications after June 2020. Thus, we intentionally focus on the literature that existed during the early stages of the COVID-19 pandemic, when healthcare settings made emergency response decisions about PPE.

Contrary to our initial hypothesis, the average HDI of study locations included in this review was higher than the worldwide average. Only two included studies attempted to summarize the diverse field of PPE use and conservation practices in low-resource settings [33,64]. Off-label reuse practices frequently occur in settings without strict regulations [33], but the safety and efficacy of these practices have not been studied. Further research on these practices would illuminate both safer approaches to resource conservation in resource-limited settings and safe PPE conservation practices that can be employed broadly for greater environmental sustainability and resilience [19].

The methods deemed successful by this body of research cannot be generalized to low- and middle-income countries, which might have different resource and cost constraints [82]. Future literature reviews are needed to address dimensions of PPE use and reuse that were not evaluated in our systematic review, such as cost and comfort.

## 5. Conclusions

This review identified multiple methods for safe and effective reuse and disinfection of single-use PPE such as N95 respirators, which can be viable options for conserving PPE in different healthcare settings and for use by the public. It also highlights that surgical masks, which are lower in cost compared with N95 respirators, are effective in reducing the risk of infection from pathogens spread via respiratory droplets and fomites.

Future research should prioritize (i) assessing the type and effectiveness of PPE conservation practices in developing contexts, (ii) the development of low-cost PPE that can be disinfected and reused, (iii) the creation of protocols for healthcare systems to safely reuse and disinfect existing single-use PPE during times of extreme shortage, and (iv) the standardization of updated criteria for the use of surgical masks vs N95 respirators for specific infectious diseases. Such interventions would both bolster our preparedness for future supply shortages in times of healthcare overutilization, as well as reduce the environmentally harmful volume of waste generated by healthcare systems.

## Figures and Tables

**Figure 1 ijerph-20-02575-f001:**
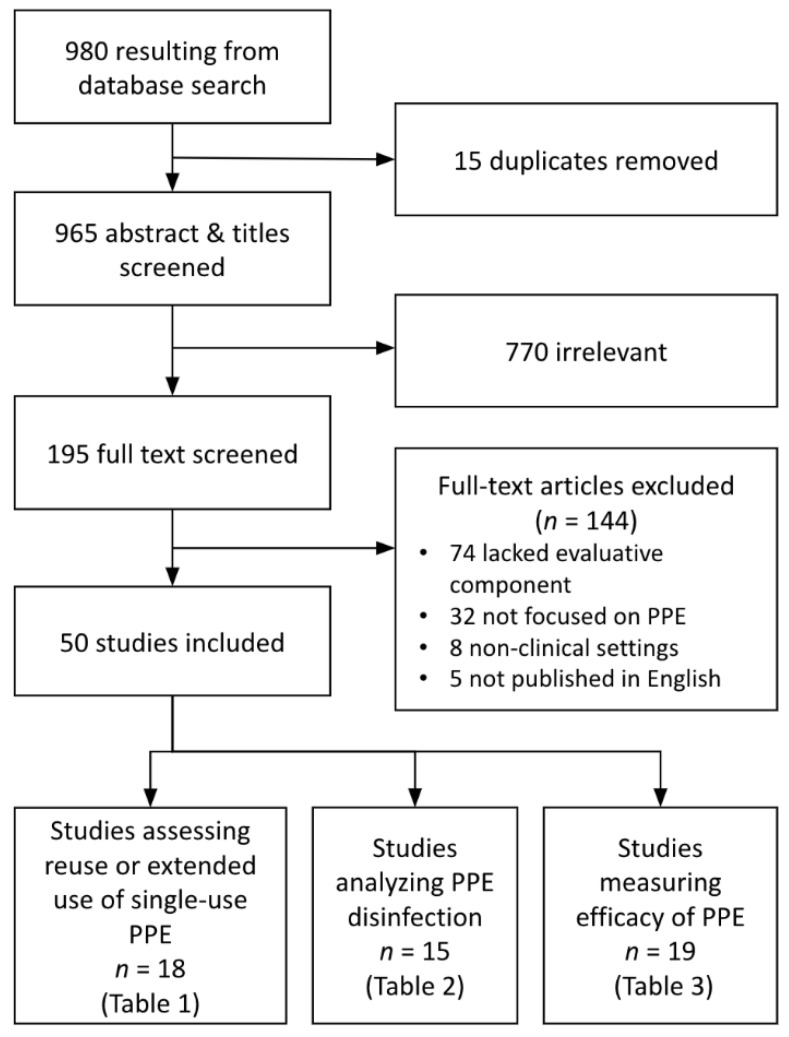
PRISMA Diagram showing literature screening process and initial sorting for analysis.

**Table 1 ijerph-20-02575-t001:** Extended use or multiple use of single-use PPE.

Study (Author, Year)	Type of PPE	Metric to Quantify PPE Efficacy (e.g., Fit Testing)	Recommended Maximum Number of Reuses (Before Non-Efficacious)	Reuse Methodology Studied	Limiting Factor(s) to PPE Reuse	Study Results Support or Are in Favor or Reuse
**Clinical In-Situ**						
**Filtration factor as metric**						
*Practices around the use of masks and respirators among hospital health care workers in 3 diverse populations* (Chughtai 2015 [33])	Surgical masks; Cloth masks; N95 Respirators	Filtration factor, observation of type of PPE used in different countries and disease risk groups	During shortages, N95 respirators can be reused until they are visibly soiled or damaged. Respirators are more effective than improvised masks or unproven decontamination methods.	Observation of healthcare workers’ reuse of PPE over multiple shifts during shortages, practiced without policies regulating this.	Respirators must not be visibly soiled or damaged. Disinfection can degrade masks and respirators. Reusing cloth masks can increase the risk of contamination from pathogens.	Yes
**Contamination as metric**						
*Absence of contamination of personal protective equipment (PPE) by severe acute respiratory syndrome coronavirus 2 (SARS-CoV-2)* (Ong 2020 [30])	N95 respirators; Eyewear (goggles); Shoes	Viral load of coronavirus contamination on PPE of healthcare workers	No limit was found; no coronavirus contamination was found on surfaces of PPE after contact with patients in a clinical setting.	Extended wear, swabbing for virus after contact with patients for up to 25 min	Insufficient environmental factors (such as air circulation) and hand hygiene	Yes
*Reusable protective eyewear tied to greater risk of contamination* (Mathias 2015 [34])	Reusable eyewear; Disposable eyewear	Contamination after use	Reusable protective eyewear is not recommended. Disinfection was unsuccessful and contamination remained.	Reusable protective eyewear wiped with germicidal wipes, compared with single-use disposable eyewear	Protective eyewear is difficult to disinfect and can harbor contaminants.	No
**Degradation of PPE as metric**						
*Maintenance status of N95 respirator masks after use in a health care setting.* (Duarte 2010 [35])	N95 respirators	Frequency of folds, dirt, and stains found on masks after nursing shifts	Respirators should be used only for shifts less than 12 h but could be used up to 5 days in a row if not visibly soiled.	Extended use for the duration of 6 or 12 h shifts, for up to 30 days	Stains, folds, and dirt appeared on respirators after all nursing shifts (regardless of length) and could alter fit or protection level	Yes
*Disinfection of reusable elastomeric respirators by health care workers: A feasibility study and development of standard operating procedures* (Bessesen 2015 [36])	Reusable elastomeric respirators	Stretching of respirator strap; adherence to proper disinfection procedures by staff	45 days of reuse after daily treatment with the disinfection process	Disinfection Standard Operating Procedure (SOP): disassemble respirator, wipe filters with disinfectant wipes, clean mask with bleach disinfecting solution, dry overnight	Respirator straps stretched 0–7.1% longer than baseline, depending on the model, after 45 days of daily disinfection treatment	Yes
**Lab-Based**						
**Filtration Factor as metric**						
*Impact of multiple consecutive donnings on filtering facepiece respirator fit* (Bergman 2012 [37])	Respirators	Filtration factor (FF greater than 100 was considered effective protection)	Up to 5 uses. Although some samples performed through 20 donnings, the study concludes that 5 consecutive donnings can be consistently performed on a N95 filtering facepiece respirator before fit factors drop below 100.	Repeated donning and doffing	Fit test failures caused by greater stress on respirator components such as head straps	Yes (up to 5 times)
**Contamination as metric**						
*Simulation as a tool for assessing and evolving your current personal protective equipment: lessons learned during the coronavirus disease (COVID-19) pandemic* (Lockhart 2020 [38])	Gowns (reusable)	Contamination through PPE, simulated particles visualized with UV light	None: switched from reusable to disposable gowns after simulation	Careful doffing. Original recommended PPE was modified to: disposable surgical gown, double high-cuffed gloves, surgical hood with ties, and shoe covering to prevent soilage and contamination	Self-contamination via doffing; contamination from liquid soilage through permeable reusable gowns	No
*Evaluation of the Survivability of Microorganisms Deposited on Filtering Respiratory Protective Devices under Varying Conditions of Humidity* (Majchrzycka 2016 [39])	N95 respirators/HEPA filters	Survivability of microorganisms on filter materials	Safe reuse is dependent on storage conditions for PPE (requires limited humidity to prevent survival of contaminating microorganisms)	Environmental conditions and materials	Survival of bioaerosols in humid conditions and on particular filter materials	Yes
*Transfer of bacteriophage MS2 and fluorescein from N95 filtering facepiece respirators to hands: Measuring fomite potential* (Brady 2017 [40])	N95 respirators	Viral load from contamination on PPE and user from repeated donning and doffing	Dependent on proper doffing and reuse technique	Observed transfer of MS2 and fluorescein after improper versus proper doffing	User error: improper doffing technique can cause self-contamination	Yes (if proper technique used)
**Degradation of PPE as metric**						
*Impact of three biological decontamination methods on filtering facepiece respirator fit, odor, comfort, and donning ease* (Viscusi 2011 [37])	N95/Filtering Facepiece Respirators	Filtration factor; reduction in fit; odor, comfort, and donning ease post-decontamination	Indefinite with UVGI, moist heat, or microwave-generated steam disinfection: decontamination methods did not reduce efficacy or comfort	3 types of disinfection: ultraviolet germicidal irradiation; moist heat incubation; microwave-generated steam	Individual models of PPE have different results	Yes
*Analysis of forces generated by N95 filtering facepiece respirator tethering devices: a pilot study* (Roberge 2012 [41])	N95 respirators	Load/forces on the respirator tethering device	2 donnings. There was a progressive decline in loads generated by tethering devices over the course of multiple donning and doffing episodes, but the load decrement for donnings 1 and 2 were not significant.	Repeated donning and doffing, using a tethering device for added comfort and protection	Load and stress on tethering devices from repeated use	Yes (up to 2 times)
*Evaluation of Five Decontamination Methods for Filtering Facepiece Respirators* (Viscusi 2009 [42])	N95/Filtering Facepiece Respirators	Filter aerosol penetration and resistance before and after disinfection	Indefinite: UVGI, ethylene oxide, vaporized hydrogen peroxide, and bleach were effective for prolonging use	Disinfection: ultraviolet germicidal irradiation (UVGI), ethylene oxide, vaporized hydrogen peroxide (VHP), microwave oven irradiation, or bleach	Microwave irradiation made FFRs unwearable after treatment due to melting	Yes
*Effects of Ultraviolet Germicidal Irradiation (UVGI) on N95 Respirator Filtration Performance and Structural Integrity* (Lindsley 2015 [43])	N95 respirators	Penetration; flow resistance; strength of respirator coupon & straps	Each individual model of N95 should be tested separately to determine whether UVGI can be done safely	UV germicidal irradiation	High doses of UVGI caused layers of the respirator to lose strength and reduce efficacy	No
*Analysis of residual chemicals on filtering facepiece respirators after decontamination* (Salter 2010 [44])	N95/Filtering Facepiece Respirators	Residues on FFR from disinfection	Indefinite: hydrogen peroxide, vaporized hydrogen peroxide, and ultraviolet light can effectively enable reuse.	Disinfection: hydrogen peroxide, sodium hypochlorite, mixed oxidants, dimethyldioxirane, ethylene oxide, vaporized hydrogen peroxide, ultraviolet light.	Ethylene oxide left toxic residues, and some methods caused degradation: bleach left an odor, corroded metal, and caused some discoloration; DMDO and Mixed oxidants left odors and oxidation on metal.	Yes
*Sterilization of disposable face masks by means of standardized dry and steam sterilization processes; an alternative in the fight against mask shortages due to COVID-19* (de Man 2020 [45])	N95/Filtering Facepiece Respirators	Pressure drop over masks; mask permeability; Filtration factor; pressure/flow and particle tests; bacterial spray penetrance	Indefinite: masks can be sterilized multiple times and be reused safely. Multiple heat sterilization doses did not alter the efficacy of masks.	Disinfection by steam or dry sterilization processes (steam sterilization of used masks at 121 °C in laminated bags)	Only 3M^®®^ masks were studied	Yes
*Can N95 Respirators Be Reused after Disinfection? How Many Times?* (Liao 2020 [46])	N95 Respirators- Melt-blown fabric	Filtration efficiency; number of treatment cycles before non-efficacious; limits of temperature and humidity; pressure drop	Heat disinfection, followed by UV radiation, was the most effective method for preserving filtration properties in melt-blown fabrics & N95 respirators.	Five types of disinfection: heat under various humidities; steam; alcohol; bleach; ultraviolet germicidal irradiation	Melt-blown fabric was tested instead of complete masks. Disinfection with liquids and vapors had high risk of damaging the filters’ function.	Yes
*A method to determine the available UV-C dose for thedecontamination of filtering facepiece respirators* (Fisher 2011 [47])	N95 Filtering Facepiece Respirators	UV-C irradiance on layers to measure effective decontamination	UV-C can effectively decontaminate and prolong use of FFRs with proper dosage for transmittance	UV-C decontamination	UV-C cannot always penetrate multiple layers and porous surfaces of respirators for decontamination unless the proper dose of UV-C is used	Yes
*Reusable and Recyclable Graphene Masks with Outstanding Superhydrophobic and Photothermal Performances* (Zhong 2020 [48])	Novel PPE—surgical masks with graphene coating	N/A, Self-heats with exposure to sun for disinfection and surface is hydrophobic.	Indefinite: covering masks in a layer of graphene enabled reuse of surgical masks by high-temperature self-cleaning from solar illumination.	Solar illumination (temperature over 70C when graphite exposed to solar illumination)	Filtration efficacy of modified PPE not studied	Yes

**Table 2 ijerph-20-02575-t002:** PPE disinfection methods against pathogens.

Study (Author, Year)	Type of PPE	Pathogen	Study Design	Disinfection Method(s) Studied	Metric(s) to Quantify Efficacy	Recommendations
**Bacterial**						
*Test methods for estimating the efficacy of the fast-acting disinfectant peracetic acid on surfaces of personal protective equipment* (Lemmer 2017 [53])	Gowns; protective eyewear (shields, goggles); gloves; protective boots	Bacillus subtilis; vaccinia virus; adenovirus	Lab-based	Peracetic acid (PAA) tested by submerging PPE in PAA solution, by covering PPE surface with PAA solution, and by spraying PAA solution on PPE.	Spore reduction factor, inactivation of viruses	0.5–1% Peracetic acid (PAA) inactivated Vaccinia virus and Adenovirus by 6 log10. Combining PAA with detergent killed *B. subtilis* on hydrophobic PPE surfaces, but PAA alone was not effective against *B. subtilis*.
*It’s not the heat, it’s the humidity: Effectiveness of a rice cooker-steamer for decontamination of cloth and surgical face masks and N95 respirators* (Li 2020 [54])	Surgical masks; N95/respirators/elastomeric/HEPA; cloth	MRSA (methicillin-resistant Staphylococcus aureus) and MS2 (single-stranded RNA virus bacteriophage)	Lab-based	2 types: (1) steaming—cycle of treatment in rice cooker-steamer (13–15 min); (2) dry heat—oven at 100 °C (15 min)	Reduction in presence of inoculated organisms (efficacy threshold minimum of 3log10 reduction in viable MS2 or MRSA)	Steam treatment in a rice cooker-steamer had >5 log 10 reduction in MS2 and MRSA. Dry heat at the same temperature levels was much less effective.
*Evaluation of Sporicidal Disinfectants for the Disinfection of Personal Protective Equipment During Biological Hazards* (Papp 2020 [57])	Gown material	Bacterial spores	Lab-based	3 active ingredients: chlorine-, peracetic acid-, and oxygen-releasing disinfectants	Spore reduction factor; number of remaining viable spores after 48 h	2% chlorine-based disinfection and 1.75% peracetic acid significantly reduced bacterial spores on PPE. Oxygen-based disinfectants were less effective.
*Reusable protective eyewear tied to greater risk of contamination* (Mathias 2015 [34])	Eyewear (shields, goggles)	Bacteria	Clinical/In-situ	Germicidal wipes	Contamination of eyewear, percentage that cultured positive for common hospital pathogens before/after attempted disinfection	After use, 37.7% of disposable eye protection and 94.9% of reusable eye protection was contaminated. After disinfection, ~75% of reusable eye protective remained contaminated. Author conclusions: operating room staff should wear disposable protective eyewear and dispose after each case.
**Viral**						
*Institution of a Novel Process for N95 Respirator Disinfection with Vaporized Hydrogen Peroxide in the setting of the COVID-19 Pandemic at a Large Academic Medical Center* (Grossman 2020 [31])	N95/respirators/elastomeric/HEPA	SARS-CoV-2	Clinical/In-situ	Vaporized hydrogen peroxide	Type of bags holding N95 respirators during disinfection; impact of pouch placement on disinfection effectiveness; quantitative fit testing after disinfection	Vaporized hydrogen peroxide reproducibly disinfected N95 respirators in Tyvek pouches. The process is scalable for a large academic hospital and healthcare system facing respirator shortages.
*A pandemic influenza preparedness study: Use of energetic methods to decontaminate filtering facepiece respirators contaminated with H1N1 aerosols and droplets* (Heimbuch 2011 [50])	Filtering facepiece respirators	H1N1 influenza	Lab-based	3 types of energetic methods: (1) microwave generated steam, (2) warm moist heat, & (3) ultraviolet germicidal irradiation.	Average reduction of viable H1N1 influenza on FFRs against both droplet and aerosol challenges per disinfection method; percent of FFRs in which virus became undetectable.	All three methods reduced viable H1N1 virus by >4 log. In 93% of experiments, no detectable virus was detected after disinfection.
*Disinfecting personal protective equipment with pulsed xenon ultraviolet as a risk mitigation strategy for health care workers* (Jinadatha 2015 [51])	Plastic face shield, gown material, glass carriers	Canine parvovirus (surrogate for Ebola)	Lab-based	Pulsed xenon ultraviolet (PX-UV)	Effectiveness of PX-UV disinfection on different surfaces measured by viral culture; amount of UV that penetrates PPE material when 1 m away from UV source	A >4 log virus reduction was found on inoculated glass, face shield and gown materials. UV light penetrance did not exceed safety limits for exposure.
*Assessment of half-mask elastomeric respirator and powered air-purifying respirator reprocessing for an influenza pandemic* (Lawrence 2017 [52])	Half-mask elastomeric respirator (HMER) and powered air purifying respirator (PAPR) masks	H1N1 influenza	Lab-based	2 types: cleaned with detergent and water +/− subsequent disinfection with bleach soak	Viral load in the presence of artificial skin oil; log reduction and viable influenza recovery after reprocessing	Cleaning with detergent significantly reduced recoverable virus, except for the strap of one model of HMER. No significant difference seen between cleaning only vs cleaning followed by bleach soak.
*Effectiveness of three decontamination treatments against influenza virus applied to filtering facepiece respirators* (Lore 2012 [55])	N95/respirators/elastomeric/HEPA	General influenza	Lab-based	3 types: ultraviolet germicidal irradiation (UVGI), microwave-generated steam (MGS), moist heat (MH)	Viral load by culture after disinfection	UVGI, MGS, & MH were effective decontamination treatments for 3M^®®^ 1860 and 1870 respirators, meeting EPA criteria for virucidal test effectiveness.
*Ultraviolet germicidal irradiation of influenza-contaminated N95 filtering facepiece respirators* (Mills 2018 [56])	N95/respirators/elastomeric/HEPA	H1N1 influenza	Lab-based	Ultraviolet germicidal irradiation	Mean viable influenza recovered from respirator surfaces alone and respirators contaminated with surrogates for saliva and skin oil	One minute of UVGI exposure significantly reduced viable influenza on respirator facepieces and straps. This varied by model, but UVGI may be an efficient and rapid method scalable for hospitals during an influenza pandemic.
*Preventing Viral Contamination: Effects of Wipe and Spray-based Decontamination of Gloves and Gowns* (Robinson 2019 [58])	Gowns; gloves	MS2 (single-stranded RNA virus bacteriophage)	Lab-based	Bleach wipe or spray on PPE worn by manikin	Quantification of plaque forming units post-decontamination); reduction in viral contamination	Bleach solution spray and wipes effectively reduced MS2 contamination on gowns and gloves, and none was detected on nearby environmental surfaces.
*Disinfection of N95 respirators by ionized hydrogen peroxide in pandemic coronavirus disease 2019 (COVID-19) due to SARS-CoV-2* (Cheng 2020 [60])	N95 respirators	H1N1 influenza	Lab-based	7.8% H_2_O_2_ solution, converted to ionized H_2_O_2_ after cold plasma arc, contact with N95 respirator in gas form	Presence of virus after treatment	Ionized hydrogen peroxide effectively inactivated influenza virus on N95 respirators
*Evaluation of Microwave Steam Bags for the Decontamination of Filtering Facepiece Respirators* (Fisher 2011 [47])	N95/respirators/elastomeric/HEPA	MS2 (single-stranded RNA virus bacteriophage)	Lab-based	Microwave steam bags	Decontamination efficacy; filtration efficiency; water absorption in respirators	Steam bags inactivated 99.9% of MS2, and all respirators maintained 95% filtration efficiency, but decontamination absorption was model-specific. More studies are required before reuse can be endorsed.
*Decontamination of face masks with steam for mask reuse in fighting the pandemic COVID-19: experimental supports* (Ma 2020 [61])	Surgical masks; N95/respirators/elastomeric/HEPA	Avian coronavirus of infectious bronchitis	Lab-based	Steam	Filtration factor; quantitative presence of active avian coronavirus after 3 days incubation	Steaming for at least 5 min with boiling water inactivated avian coronavirus. Steaming for up to 120 min did not reduce the filtration efficiency of masks or respirators and masks could be reused for 7–10 days if undamaged, clean and with good fit.
*Effectiveness of common healthcare disinfectants against H1N1 influenza virus on reusable elastomeric respirators* (Subhash 2014 [62])	N95/respirators/elastomeric/HEPA	H1N1 influenza	Lab-based	3 types of disinfectant wipes: (1) 70% isopropyl alcohol, (2) quaternary ammonium chloride (QAC) plus 17.2% isopropyl alcohol, (3) 1: 10 bleach dilution plus detergent	Viral load after disinfection, by culture and by PCR	QAC/isopropyl alcohol and bleach with detergent disinfectant wipes effectively disinfected H1N1 influenza from elastomeric respirator material. However, 62.5% of samples treated with bleach plus detergent had virus detected by PCR, warranting further study.

**Table 3 ijerph-20-02575-t003:** Efficacy evaluations of PPE.

Pathogen	Study (Author, Year)	Type(s) of PPE Studied	Metric for Efficacy	Clinical/In Situ vs. Lab-Based	Conclusions
**PPE efficacy—measured by infection rate.**					
**Pathogen Type: Virus. Predicted Mode of Transmission: Droplets and Aerosols**					
SARS-COV-2	*Association between 2019-nCoV transmission and N95 respirator use* (Wang 2020 [32])	N95	Infection rate	Clinical (observational)	N95 respirators significantly reduced the risk of SARS-COV-2 in healthcare workers, with 0% infection for the N95 group compared with 4.6% in those with no mask.
General respiratory illness	*A Randomized Clinical Trial of Three Options for N95 Respirators and Medical Masks in Health Workers* (MacIntyre 2013 [49])	Surgical masks and N95 respirators, targeted use vs continuous use	Infection rate of clinical respiratory illness, influenza-like illness, laboratory-confirmed respiratory virus, laboratory-confirmed influenza	Clinical (randomized)	Healthcare workers with continuous use of N95s had lower rates of clinical respiratory illness but not other respiratory viral diagnoses compared with continuous use of medical masks and targeted use of N95s.
General respiratory illness	*A cluster randomized clinical trial comparing fit-tested and non-fit-tested N95 respirators to medical masks to prevent respiratory virus infection in health care workers* (MacIntyre 2011 [63])	Surgical masks and N95 respirators	Infection rate of clinical respiratory illness, influenza-like illness, laboratory-confirmed respiratory virus, and influenza.	Clinical (randomized)	Healthcare workers wearing N95s had lower rates of clinical respiratory illness than those wearing surgical masks. No difference in any other metric.
General respiratory illness	*A cluster randomised trial of clothmasks compared with medical masks in healthcare workers* (MacIntyre 2015 [64])	Cloth and surgical masks	Infection rate of clinical respiratory illness and influenza-like illness; Penetration by particles	Clinical (randomized)	Surgical masks are more effective than cloth masks in preventing clinical respiratory illness
General respiratory illness	*Cost-effectiveness analysis of N95 respirators and medical masks to protect healthcare workers in China from respiratory infections* (Mukerji 2017 [65])	Surgical masks and N95s	Rates of clinical respiratory illness	Clinical (randomized)	Clinical respiratory illness rates were higher for healthcare workers wearing surgical masks compared with N95 respirators.
General influenza virus	*Surgical Mask vs N95 Respirator for Preventing Influenza Among Health Care Workers* (Loeb 2009 [66])	Surgical masks and N95 respirators	Infection rate	Clinical (randomized)	No significant difference in influenza infection rate between healthcare workers wearing surgical masks and N95s respirators
General respiratory illness	*Use of surgical face masks to reduce the incidence of the common cold among health care workers in Japan: a randomized controlled trial* (Jacobs 2009 [67])	Surgical masks	Infection rate	Clinical (randomized)	Surgical mask use in health care workers did not reduce incidence of the common cold
General respiratory illness	*Preliminary Findings of a Randomized Trial of Non- Pharmaceutical Interventions to Prevent Influenza Transmission in Households* (Cowling 2008 [68])	Surgical masks	Infection rate	Clinical (randomized)	No significant difference in household transmission of influenza between no mask, surgical mask, and hand hygiene
Influenza H1N1	*Surgical Masks for Protection of Health Care Personnel against Pandemic Novel Swine-Origin Influenza A (H1N1)–2009: Results from an Observational Study* (Ang 2010 [69])	Surgical masks and N95 respirators	Infection rate	Clinical (observational)	No significant difference in influenza H1N1 infection rate between healthcare workers wearing surgical masks and N95 respirators
General influenza virus	*N95 Respirators vs Medical Masks for Preventing Influenza Among Health Care Personnel A Randomized Clinical Trial* (Radonovich 2019 [70])	Surgical masks and N95 respirators	Incidence of laboratory-confirmed influenza	Clinical (randomized)	No significant difference in laboratory-confirmed influenza virus between healthcare workers wearing surgical masks and N95 respirators
**Pathogen Type: Bacteria. Predicted Mode of Transmission: Fomites**					
General	*Efficacy of face masks and respirators in preventing upper respiratory tract bacterial colonization and co-infection in hospital healthcare workers* (MacIntyre 2014 [71])	Surgical masks and N95 respirators	Rate of bacterial colonization of upper respiratory tract	Clinical (randomized)	Compared with control and mask groups, N95 respirators were significantly more protective from bacterial colonization
**PPE efficacy—measured by filtration factor**					
**Pathogen Type: Bacteria. Predicted Mode of Transmission: Fomites**					
General	*Aspects of Tests and Assessment of Filtering Materials Used for Respiratory Protection Against Bioaerosols* (Majchrycka 2010 [72])	Nonwovens (antimicrobial textile filter material)	Filtration factor	Lab	Biocidally active non-woven filter material are active against bacteria deposited on the surface (*E. coli* and *S. aureus*) and had filtration efficiency of 86–95%.
**Pathogen Type: Virus. Predicted Mode of Transmission: Droplets and Aerosols**					
Avian influenza virus	*Potential utilities of mask-wearing and instant hand hygiene for fighting SARS-CoV-2* (Ma 2020 [61])	Surgical masks; N95 respirators; homemade masks (four-layer kitchen paper towels and one-layer cloth)	Percent of aerosols blocked	Lab	Homemade cloth masks, disposable N95s and surgical masks blocked >95% of aerosols. Layers of paper towel can make homemade masks effective, and replacing paper towels makes it easily reusable.
**Pathogen Type: Non-disease specific**					
	*Pretreated household materials carry similar filtration protection against pathogens when compared with surgical masks* (Carnino 2020 [73])	Paper towel and surgical mask material treated with NaCl, untreated surgical mask material	Filtration factor	Lab	Paper towels and surgical masks treated with a saline solution are more effective in filtering out small particles and bacteria compared with untreated mask material.
	*A randomised controlled pilot study to compare filtration factor of a novel non-fit-tested high-efficiency particulate air (HEPA) filtering facemask with a fit-tested N95 mask* (Au 2010 [74])	Non-fit-tested HEPA mask and fit-tested N95	Filtration factor	Lab	Fit-tested N95 was more effective than the non-fit-tested HEPA mask. Authors do not recommend the HEPA mask without fit testing.
**PPE efficacy—other metric**					
**Pathogen Type: Virus. Predicted Mode of Transmission: Droplets and Aerosols**					
Influenza H1N3	*Universal and reusable virus deactivation system for respiratory protection* (Quan 2017 [75])	Surgical mask material treated with sodium chloride	Infection rate in mice; filtration efficiency; viral load and inactivation from exposure to salt	Lab	A coating of sodium chloride increases filtration efficiency of masks and inactivates pathogens from aerosols to prevent infectivity.
**Pathogen Type: Non-disease specific**					
	*Professional and home-made face masks reduce exposure to respiratory infections among the general population* (van der Sande 2008 [76])	Filtering facepiece masks (FFP2), surgical masks, homemade masks	measurement of aerosol particles	Lab	FFP2 were most effective in blocking aerosols, followed by surgical masks, and then homemade masks
	*Safety testing improvised COVID-19 personal protective equipment based on a modified full-face snorkel mask* (Greig 2020 [77])	Full-face snorkel mask (novel PPE)	Fit testing	Lab	Novel mask failed quantitative fit testing. Authors do not recommend using novel PPE without quantitative filtration testing
	*Simulation as a tool for assessing and evolving your current personal protective equipment: lessons learned during the coronavirus disease (COVID-19) pandemic* (Lockhart 2020 [38])	Surgical masks, N95s, gowns, eyewear	Permeability of PPE under study to liquid, risk of self-contamination	Lab	Contamination beneath protective gowns is highly possible even with meticulous donning and doffing

## Data Availability

A spreadsheet of included articles and extracted data is available upon request.

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
