# Peer review of "Conservation Practices for Personal Protective Equipment: A Systematic Review with Focus on Lower-Income Countries"

_ijerph, 2023, doi:10.3390/ijerph20032575_

Round 1

Reviewer 1 Report

 Dear Authors,

It was a pleasure reading your well-structured review. Few places the same idea has been repeated which adds to the redundancy, which is recommended to be omitted. Furthermore, I found that surface disinfectants solutions were not emphasized. What do you mean by eye glasses? Face shields you mean? The acrylic /plastic transparent covers or what exactly? 

The below part is not edited as per your paper:

Institutional Review Board Statement:  In this section, you should add the Institutional Review  

Board Statement and approval number, if relevant to your study. You might choose to exclude this statement if the study did not require ethical approval. Please note that the Editorial Office might  ask you for further information. Please add “The study was conducted in accordance with the Dec

laration  of  Helsinki,  and  approved  by  the  Institutional  Review  Board  (or  Ethics  Committee)  of NAME OF INSTITUTE (protocol code XXX and date of approval).” for studies  involving humans.  

OR “The animal study protocol  was approved by the Institutional Review Board (or Ethics Committee) of NAME OF INSTITUTE (protocol code XXX and date of approval).” for studies involving animals. OR “Ethical review and approval were waived for this study due to REASON (please provide a detailed justification).” OR “Not applicable” for studies not involving humans or animals. 

Informed Consent Statement:  Any research article describing a study involving humans should contain this statement. Please add “Informed consent was obtained from all subjects involved in the study.” OR “Patient consent was waived due to REASON (please provide a detailed justification).” OR “Not applicable.” for studies not involving humans. You might also choose to exclude this statement if the study did not involve humans. Written informed consent for publication must be obtained from participating patients who can be identified (including by the patients themselves). Please state “Written informed consent has been obtained from the patient(s) to publish this paper”  if applicable.

If not mention NOT APPLICABLE. 

The changes in the entry and exit of positive patients would be another nice criterion to compare, as many low-income countries were having small advantages of separation of units, and saving the spread to the non-infected admitted patients.

Author Response

We have taken into account your thoughtful comments and suggested edits. Specifically:
1. We have addressed potential areas of repetition to reduce redundancy in the Results and Discussion.
2. We have added more information in the tables describing surface disinfectants evaluated in the studies.
3. We have also clarified in the tables and text our descriptions of eye protection, because some studies evaluated different forms of eye protection (goggles and face shields were the most common type of eye protection included).
4. We have also added a statement at the end of our manuscript to address the ethical conduct of research, and that our study was exempt from IRB review because it did not include human subjects research.
5. Lastly, we did not find any publications that met our inclusion criteria published during the period leading up to June 2020 that compared changes in the entry and exit of positive patients, so we are unable to address your final comment. We agree that this aspect of infection prevention would benefit from further study.

Reviewer 2 Report

The authors of this paper made a very good exercise to clarify PPE practices. Even accepting that results are not applicable to low-medium income countries (L-MIC), the analyses and results shown are clearly useful in the first instance to high-income countries but are leading traces for future studies in L-MIC.

I would like to suggest the authors show the "n" for each type of the ending studies in Figure 1, and include a brief commentary about it in the results section.

Author Response

We have made edits based on your helpful comments. Specifically:
1. We have added additional information to Figure 1, which now includes the number of studies for each study category (reuse and extended use of PPE, PPE disinfection methods, and efficacy of PPE) and included clarifying information regarding these numbers in our Results.

Reviewer 3 Report

Brief Summary

This is a literature review to evaluate approaches for conserving single-use disposable PPE for repeated use. This practice, if doable and successful can assist with stretching supplies of PPE when there is a shortage and provides a mechanism to reduce costs in countries and health care settings where resources are limited. Fifty one published studies from 14 countries were included in this assessment.

The authors of this review found that there are safe and effective ways to reuse N95s when standardized processes for donning and doffing and stable storage conditions exist. Improper doffing can enable the transfer of contaminants and storage conditions with limited humidity prevent the survival of contaminating microorganisms. 

Cloth and surgical masks were inferior to N95s for aerosolized organisms, while surgical masks did OK with fomites and large droplets. Of concern are the variability of materials used and fit for homemade cloth masks.

Effective disinfection methods may not be financially and/or logistically feasible during outbreaks. Methods that require significant infrastructure or cost include vaporized hydrogen peroxide and pulsed xenon ultraviolet light. In contrast, effective disinfectants using common and inexpensive methods such a microwave steam bags and steam sterilization using a kitchen rice cooker can be available in health care settings with limited resources.

The conclusion showed that more research is required to develop low-cost PPE that can be easily disinfected and reused.

Author Response

Based on your comments, there were no additional specific edits to be addressed.
My coauthors and I are thrilled our manuscript has been accepted with minor revisions. Thank you again for your time and efforts.